# Groundwater Quality and Potential Human Health Risk Assessment for Drinking and Irrigation Purposes: A Case Study in the Semiarid Region of North China

**Feifei Chen** [1] , **Leihua Yao** [1,*], **Gang Mei** [1] , **Yinsheng Shang** [2], **Fansheng Xiong** [3] **and Zhenbin Ding** [1]

[1] School of Engineering and Technology, China University of Geosciences (Beijing), Xueyuan Road 29, Beijing 100083, China; cff@cugb.edu.cn (F.C.); gang.mei@cugb.edu.cn (G.M.); dzb0607@cugb.edu.cn (Z.D.)
[2] Shanxi Survey Design Research Institute Co., Ltd., Taiyuan 030012, China; sys426@sohu.com
[3] Institute of Applied Physics and Computational Mathematics, Beijing 100094, China; hxw1334@126.com
* Correspondence: yaolh@cugb.edu.cn

**Abstract:** Groundwater is a valuable water source for drinking and irrigation purposes in semiarid regions. Groundwater pollution may affect human health if it is not pretreated and provided for human use. This study investigated the hydrochemical characteristics driving groundwater quality for drinking and irrigation purposes and potential human health risks in the Xinzhou Basin, Shanxi Province, North China. More specifically, we first investigated hydrochemical characteristics using a descriptive statistical analysis method. We then classified the hydrochemical types and analyzed the evolution mechanisms of groundwater using Piper and Gibbs diagrams. Finally, we appraised the groundwater quality for drinking and irrigation purposes using the entropy water quality index (EWQI). We assessed the associated human health risks for different age and sex groups through drinking intake and dermal contact pathways. Overall, we found that (1) Ca-HCO$_3$ and Ca·Mg-HCO$_3$ were the dominant hydrochemical types and were mainly governed by rock weathering and water–rock interactions. (2) Based on the EWQI classifications, 67.74% of the groundwater samples were classified as medium quality and acceptable for drinking purpose. According to the values of sodium adsorption ratio (SAR), residual sodium carbonate (RSC) and soluble sodium percentage (%Na), 90.32% of the samples were suitable for irrigation, while the remaining samples were unfit for irrigation because of the high salinity in the groundwater. (3) Some contaminants in the groundwater, such as NO$_3$$^-$, NO$_2$$^-$ and F$^-$, exceeded the standard limits and may cause potential risks to human health. Our work presented in this paper could establish reasonable management strategies for sustainable groundwater quality protection to protect public health.

**Keywords:** groundwater quality; nitrate; fluoride; health risk assessment; entropy water quality index; Xinzhou Basin

## 1. Introduction

Groundwater resources are indispensable for domestic drinking water supplies, irrigation resources and industrial activities, especially in arid and semiarid regions, owing to the shortage of surface water [1–3]. However, groundwater contamination has become a severe issue affecting human health and life in many countries and regions in recent years, with population growth and agriculture and industry development [4,5]. Most common contaminants in groundwater mainly include fluoride, nitrogen and many others. If they exceed the standard limits for drinking water and are released untreated for direct human use, they may have harmful effects on human health [6–9]. For example, drinking groundwater with a high nitrate concentration in the long-term can cause methemoglobinemia, gastric cancer and congenital disabilities [10,11]. Besides, long-term water consumption with fluoride concentrations exceeding the standard limits will lead to dental and skeletal fluorosis [12]. Simultaneously, pollution also affected the sustainable development of the

ecological environment and society [5,13]. Therefore, the issue of groundwater quality and safety has attracted the attention of many researchers, and groundwater quality assessments associated with health risk evaluations have received considerable amounts of attention worldwide, including in China, India and Iran [14–17].

Many traditional methods have been used to evaluate groundwater quality over the past few decades, such as fuzzy comprehensive assessment methods [18], set pair analyses [19], rough sets [20], multivariate techniques [21], hierarchical analyses [22] and water quality indexes (WQI) [23]. However, some deficiencies exist in the application of these assessment methods, such as the need to consider too many factors [24]. WQI is used to quantify water quality by weighting the importance of various evaluation indexes, widely used to describe groundwater quality due to its practicability and effectiveness [25]. However, the representation of evaluation results is sensitive to the weight of parameters [26]. Entropy weight can eliminate the influence of subjective factors on water quality parameters and give reasonable weight to the parameters, combined with WQI to effectively quantify groundwater quality [3,27,28]. In this paper, the entropy water quality index (EWQI) was used to evaluate groundwater quality.

Furthermore, health risk assessment can effectively characterize the potential impact of groundwater on human health and provide a basis for water quality management agencies to ensure water security [29]. Adimalla et al. [14] quantified the extent of the health risks associated with fluoride and nitrates in groundwater in southern India. Xiao et al. [30] appraised the quality of groundwater in the North China Plain. They found that nitrate and other pollutants have a high potential noncarcinogenic risk to the human body through groundwater ingestion. Zhang et al. [31] analyzed the groundwater quality in the Guanzhong Basin of China and the impact of nitrite, nitrate, and fluorine on the health of different populations. These researches have shown that ensuring the quality of groundwater is of great significance to the protection of human health and the sustainable development of society.

Xinzhou Basin is located in the semiarid area in central-eastern Shanxi Province, North China. Groundwater is the primary water resource in the basin, accounting for 70% of the total water consumption. Approximately 65% of the groundwater was used for irrigation and 10% for domestic use. Surface water accounted for 30% of total water consumption and came mainly from rainfall during the flood season. Although piped water networks supply urban areas, there was no piped water, and residents used untreated groundwater directly for drinking and irrigation in most rural areas. There were mainly coal, chemical, metallurgy and other industries, while agriculture was primarily corn, sorghum and vegetable cultivation in Xinzhou Basin. However, with rapid economic development since the 1980s, industrial, agricultural and mining activities have dramatically increased the water demand. From 1982 to 2000 in the Xinzhou Basin, the amount of groundwater storage decreased by $2.7896 \times 10^8$ m$^3$, with an average annual decrease of $1.469 \times 10^7$ m$^3$. Excessive groundwater exploitation had resulted in a continuous decline in groundwater levels and the deterioration of groundwater quality in urbanized and agricultural areas in recent decades, threatening approximately 1.66 million people's health and safety.

In Xinzhou Basin, due to the discharge of domestic sewage and industrial wastewater and the excessive use of agricultural fertilizer, the concentration of nitrate and nitrite in groundwater is high. Besides, fluorine poisoning has become a specific disease in the region due to the high concentration of fluoride ions in the drinking water many years ago. Since the 1980s, the area has adopted methods such as improved water use and reduced fluoride. These measures have significantly reduced the incidence of fluoride poisoning among the population. Due to the pollution of nitrate, nitrite and fluoride, groundwater quality in some areas of Xinzhou Basin was low, leading to endemic diseases for residents [32]. Because of the importance of groundwater in the Xinzhou Basin, some groundwater research has been carried out. For example, Han et al. [33–35] conducted some research to identify groundwater flow systems and indicated possible water–rock interaction processes based on hydrochemical characteristics and isotopic compositions.

Although government departments have investigated groundwater quality in Xinzhou Basin, few studies have been carried out on the potential health risks of groundwater pollutants to residents in the area combined with groundwater quality.

This paper presents a case study assessing the groundwater quality used for drinking and irrigation and its potential risks to human health. We selected the Xinzhou Basin, a semiarid region, Shanxi Province, China, as the research area. By field sampling and analysis, we obtained a series of hydrochemical data for the shallow groundwater. Specifically, we first investigated the hydrochemical characteristics using descriptive statistical methods in the Xinzhou Basin. We then classified the hydrochemical types and analyzed the evolution mechanisms of groundwater using Piper and Gibbs diagrams. Finally, we appraised the groundwater quality using the EWQI. We quantified the potential health risks of nitrates, nitrites and fluorine to residents of different age and sex groups through drinking intake and dermal contact pathways. This study could be used to establish reasonable management strategies for sustainable, groundwater quality protection to protect public health.

In this paper, our work can be described as the following aspects:

(1) We investigated the hydrochemical characteristics and evolutionary processes of shallow groundwater in the Xinzhou Basin.
(2) We assessed the groundwater quality used for drinking water and irrigation.
(3) We evaluated the potential risks posed by noncarcinogenic factors to human health.

The remainder of this study can be summarized as follows. The materials and methods used in this paper are described in detail in Section 2. Section 3 describes and analyses the results of investigating the hydrochemical characteristics of groundwater and assesses the groundwater quality and noncarcinogenic potential human health risks uncovered by our work. Section 4 discusses the uncertainty of the model and compares the research results with other studies. Finally, Section 5 summarizes several conclusions.

## 2. Materials and Methods

### 2.1. Brief Introduction to the Study Area

The study area is located in the central-eastern part of Shanxi Province in northern China. The Xinzhou Basin is situated within a longitudinal range of 112.22~113.95° E and a latitudinal range of 38.20~39.45° N and covers 3385.2 km² in total. The altitude ranges from 700 m to 3000 m (Figure 1). The Xinzhou Basin lies in the semiarid continental monsoon climate zone. The annual average amount of evaporation and precipitation is approximately 1600 mm and 474 mm, respectively, with an annual mean temperature of 9 °C. The yearly rainfall in Xinzhou Basin is seriously unevenly distributed, mainly from July to September, accounting for 60% to 70% of the total annual precipitation [34].

Xinzhou Basin is surrounded by mountains on all sides, such as Wutai Mountain and Yunzhongshan Mountain, with a "6" shaped concave in the middle. The terrain drops from the mountainous area to the basin, and the terrain inside the basin slopes slowly and asymmetrically to the Hutuo river spreading 65~70° NE from both sides.

Xinzhou Basin is a Cenozoic rift basin, with Wutai anticline in the east and Luliang anticline in the west. The strata lithology in the study area mainly includes magma, metamorphic rock, carbonate rock, sandstone, shale and basalt. Metamorphic rocks mainly include gneiss, amphibolite, quartzite and phyllite. The Quaternary sediments consist of alluvial and lacustrine gravel, silt and silty clay with a thickness of approximately 50~360 m. The main minerals are amphibole, biotite, apatite and calcite.

The primary recharge sources of groundwater in Xinzhou Basin are precipitation infiltration, lateral recharge in the mountainous area at the edge of the basin, surface water seepage recharge in flood season and irrigation infiltration recharge. Hutuo River is the main river in Xinzhou Basin, flowing from north to south, originated in the northern basin of Fanshi County and Dingxiang County in the southeast basin outflow. Along the way, there are the Yangyan River, Changle River, Yunzhong River and other larger tributaries. Hutuo River runs 191 km in the basin, with an average longitudinal drop of 1.3‰.

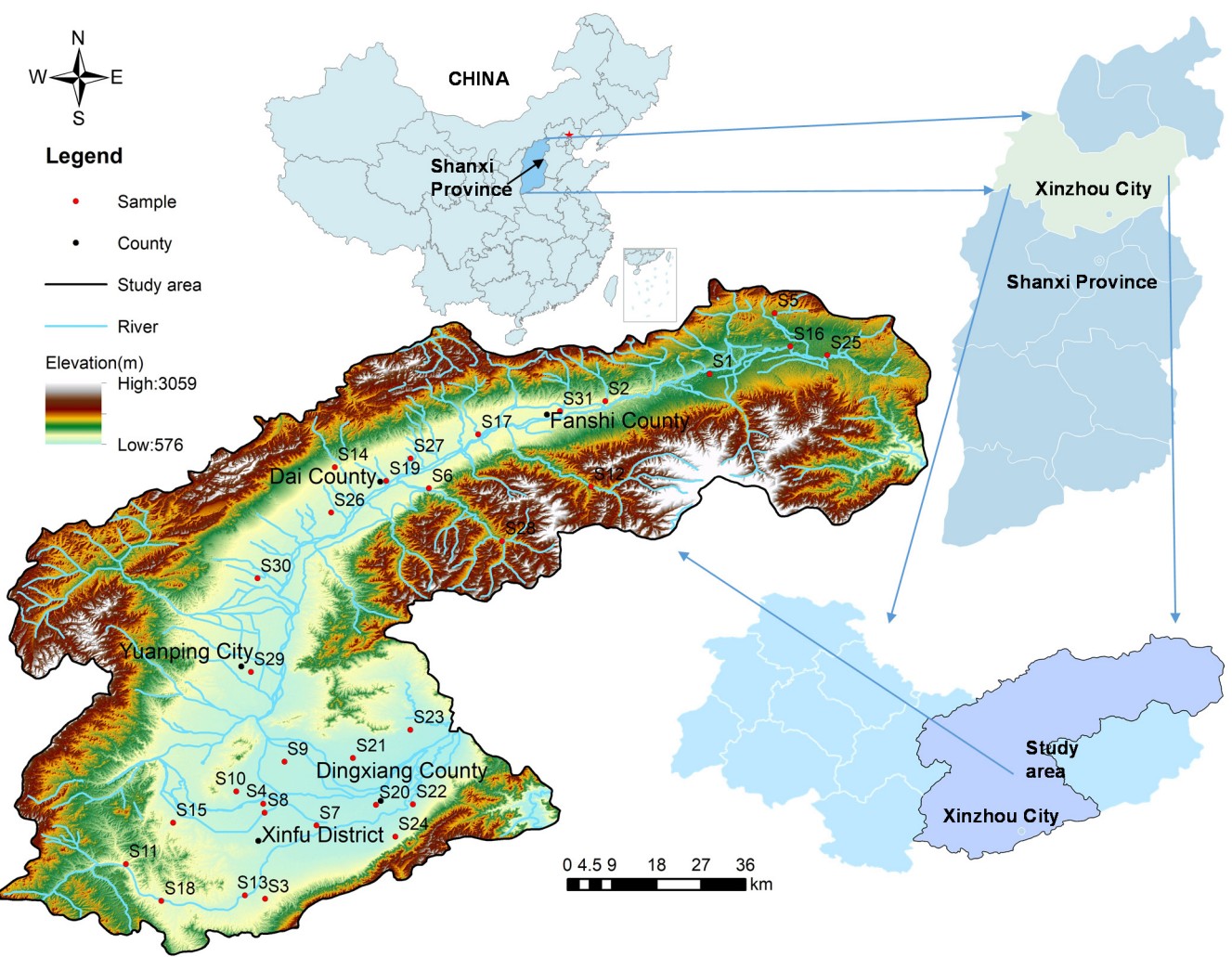

**Figure 1.** Study area and groundwater sampling sites.

*2.2. Field Sampling and Analytical Procedures*

A total of thirty-one shallow groundwater samples were collected from different wells in Xinzhou Basin, and their sampling locations were recorded by a portable global positioning system (GPS) (Figure 1). Before sampling, the wells were pumped for ten minutes to minimize the influence of residual water on the analysis results. All groundwater samples were stored in 2 L sterile polyethylene bottles. All bottles were washed two to three times using the water to be sampled before sample collection. All groundwater samples were filtered through a 0.45 μm membrane filter in the field. In order to stabilize cations in solution, groundwater samples used for cation determination were acidified to pH < 2 with concentrated nitric acid. After sampling, the groundwater samples were sealed and stored at 4 °C and transported to the laboratory immediately. Groundwater samples were collected and stored following the national technical specifications of the Ministry of Environmental Protection of the People's Republic of China [36].

Groundwater samples were analyzed in the laboratory of the Sanshui Experimental Testing Center of Shanxi Province. The groundwater quality test method referred to the Chinese Standard Test Method for Drinking Water (GB5750-2006) [37]. The following physicochemical parameters were analyzed: pH, total dissolved solids (TDS), total hardness (TH), $F^-$, $NO_3^-$, $NO_2^-$, $Na^+$, $K^+$, $Ca^{2+}$, $Mg^{2+}$, $HCO_3^-$, $SO_4^{2-}$ and $Cl^-$. TDS was determined by drying the samples at 105 °C and weighing them with an analytical balance. $Na^+$ and $K^+$ were analyzed using flame atomic absorption spectrophotometry.

$Ca^{2+}$, $Mg^{2+}$ and TH were determined by EDTA titration. $SO_4^{2-}$ and $Cl^-$ were measured using ion chromatography (ICS-90A). $HCO_3^-$ was tested using a traditional titrimetric method. $F^-$ and $NO_3^-$ were measured using ion chromatography (ICS-1500), and $NO_2^-$ was determined by spectrophotometry (7230G). Duplicates and instrument calibration were carried out for quality assurance and quality control.

The accuracy of the test results was checked by calculating the charge balance error (%CBE) for each sample by the following formula [9]:

$$\%CBE = \frac{\sum cations - \sum anions}{\sum cations + \sum anions} \times 100 \qquad (1)$$

where all cations and anions are expressed in meq/L. Generally, the %CBE should be within the range of ±5%, and the highest %CBE is 2.69% in this study, which confirms the reliability of the ion analysis results.

### 2.3. Entropy Water Quality Index (EWQI)

The WQI is a convenient evaluation method to quantify water quality that was proposed by Horton [38]. The concept of entropy was first proposed by Shannon [39]. Li et al. [40] applied the entropy weight to the traditional WQI and proposed the EWQI. The EWQI can make explicit the information of groundwater quality and eliminate the influence of human subjectivity when calculating the weight of evaluation indexes. Because of its simplicity, accuracy and consistency, the EWQI has been widely used by many scholars around the world [3,28,41–44]. In this study, the EWQI is applied to characterize the groundwater and can be calculated through six steps as follows:

**Step 1:** Establishment of the initial groundwater quality matrix. The initial matrix can be established based on the chemistry analysis data of groundwater samples. Suppose there are $m$ groundwater samples, and each sample has $n$ evaluation indexes, the initial groundwater quality matrix is X, then $x_{ij}$ is the initial value of the $j$th evaluation indicator of the $i$th groundwater sample. In this study, the values of m and n were 31 and 8, respectively.

$$X = \begin{bmatrix} x_{11} & x_{12} & \cdots & x_{1n} \\ x_{21} & x_{22} & \cdots & x_{2n} \\ \vdots & \vdots & \ddots & \vdots \\ x_{m1} & x_{m2} & \cdots & x_{mn} \end{bmatrix} \qquad (2)$$

**Step 2:** Normalization of groundwater quality matrix. There is usually a significant variation in the units and quantity grades of different groundwater quality indicators, which leads to a tremendous difference in the weight calculated. Therefore, the initial groundwater quality matrix needs to be normalized by Equation (3), where $y_{ij}$ is the normalized data value and $(r_{ij})_{min}^j$ and $(r_{ij})_{max}^j$ are the lowest and highest range of indicator $j$, respectively. Then, the standard groundwater quality matrix Y can be defined in Equation (4).

$$y_{ij} = \begin{cases} \frac{r_{ij} - (r_{ij})_{min}^j}{(r_{ij})_{max}^j - (r_{ij})_{min}^j} \text{(efficiency type)} \\ \frac{(r_{ij})_{max}^j - r_{ij}}{(r_{ij})_{max}^j - (r_{ij})_{min}^j} \text{(cost type)} \end{cases} \qquad (3)$$

$$Y = \begin{bmatrix} y_{11} & y_{12} & \cdots & y_{1n} \\ y_{21} & y_{22} & \cdots & y_{2n} \\ \vdots & \vdots & \ddots & \vdots \\ y_{m1} & y_{m2} & \cdots & y_{mn} \end{bmatrix} \qquad (4)$$

**Step 3:** Determination of information entropy. The ratio $p_{ij}$ of the index value $j$ of sample $i$ can be calculated by Equation (5), and the correction factor $10^{-4}$ is used to ensure

the significance of the equation. Hence the information entropy $e_j$ of indicator $j$ can be computed by Equation (6).

$$p_{ij} = \frac{y_{ij} + 10^{-4}}{\sum_{i=1}^{m}(y_{ij} + 10^{-4})} \tag{5}$$

$$e_j = -\frac{1}{\ln m} \sum_{i=1}^{m} p_{ij} \ln p_{ij} \tag{6}$$

**Step 4:** Calculation of entropy weight. The larger the value of $e_j$ is, the smaller the influence of the $j$ index. The entropy weight of each indicator $w_j$ can be achieved according to Equation (7).

$$w_j = -\frac{1 - e_j}{\sum_{j=1}^{n}(1 - e_j)} \tag{7}$$

**Step 5:** Determination of the quality rating scale. According to Equation (8), the quality rating scale $q_j$ of index $j$ can be determined, where $c_j$ indicates the content of index $j$ (mg/L) and $s_j$ denotes the standard limit of index $j$ (mg/L) of drinking water quality in China in this study.

$$q_j = \frac{c_j}{s_j} \times 100 \tag{8}$$

**Step 6:** Computation of EWQI. The EWQI can be computed by using Equation (9).

$$\text{EWQI} = \sum_{j=1}^{n} w_j \times q_j \tag{9}$$

According to the EWQI, the quality of groundwater for human consumption can be divided into 5 ranks, extending from extremely poor to excellent water [45], and more details are listed in Table 1.

**Table 1.** Groundwater quality classification criteria according to EWQI [45].

| EWQI | Grade | Groundwater Quality |
|------|-------|---------------------|
| <25 | 1 | Excellent |
| 25–50 | 2 | Good |
| 50–100 | 3 | Medium |
| 100–150 | 4 | Poor |
| >150 | 5 | Extremely poor |

*2.4. Human Health Risk Assessment*

Contaminated groundwater may pose negative impacts on human health via oral intake and dermal contact [31,46,47]. Long-term drinking and exposure to contaminated groundwater will seriously harm human health, which will lead to fluorosis and gastric cancer and other diseases [48]. Therefore, it is necessary to evaluate the health risks posed by pollutants on human health to provide information support for groundwater management [4,24,49]. It is useful to assess the potential harmful effects of exposure to pollutants over a certain period of time on human health [31]. The Human Health Risk Assessment (HHRA) model proposed by the United States Environmental Protection Agency (USEPA) is the most widely used model to study the potential effects of groundwater contamination on human health [6,25,50].

In this study, the health risks caused by oral and dermal contact were quantified using the evaluation model recommended by the Ministry of Environmental Protection of China [51], which is based on the USEPA model. According to different factors such as weight and daily water consumption, the exposed population is divided into three categories: children, women and men [50]. Similar to the USEPA model, the assessment process involves four steps: hazard identification, dose response assessment, exposure

assessment and risk description [24,43,52]. However, the Chinese model assigns unique parameters based on the characteristics of Chinese residents [50,53].

According to the available groundwater sample data and water quality assessment results, $NO_3^-$, $NO_2^-$ and $F^-$ were representative contaminants in this area, so they are selected as risk assessment indicators in this study. Since the contents of heavy metal ions and organics that may cause harm to the human body are not detected in groundwater, they were not calculated in the health risk assessment in this study. $NO_3^-$, $NO_2^-$ and $F^-$ are noncarcinogenic pollutants, referring to the International Agency for Research on Cancer (IARC) and USEPA, so only noncarcinogenic risks were considered in this research [53,54]. The models for noncarcinogenic risk through oral and dermal intake are as follows [6,50,53].

The noncarcinogenic risk through the oral intake is expressed as:

$$Intake_{oral} = \frac{C \times IR \times EF \times ED}{BW \times AT} \tag{10}$$

$$HQ_{oral} = \frac{Intake_{oral}}{RfD_{oral}} \tag{11}$$

where $Intake_{oral}$ is the average daily dose per unit weight by the oral pathway (mg/(kg·d)); $C$ represents the content of evaluation index (mg/L); $IR$ indicates the intake rate of groundwater (L/d) through the oral pathway. In this study, the groundwater oral intake rate for adults and children under 12 years old is 1.5 L and 0.7 L, respectively. $EF$ represents the exposure frequency (d/a), and its value is 365. $ED$ is exposure duration (a), which value for adults and children over the age of 12 is 30 and 12 years, respectively. $BW$ and $AT$ are the average weight (kg) and time (d) of noncarcinogenic effects. The weight of children, women and men is 15 kg, 55 kg and 70 kg respectively. For adults, the average duration of noncarcinogenic effects was 10,950 days, and for children, it was 4380 days. $HQ_{oral}$ and $RfD_{oral}$ represent the hazard quotient and reference dose (mg/(kg·d)) of noncarcinogenic pollutants through oral exposure. In this study, the $RfD_{oral}$ values for $F^-$, $NO_2^-$, and $NO_3^-$ were 0.04, 0.1 and 1.6 mg/(kg·d), respectively [53].

The noncarcinogenic risk through dermal contact is expressed as:

$$Intake_{dermal} = \frac{DA \times EV \times SA \times EF \times ED}{BW \times AT} \tag{12}$$

$$DA = K \times C \times t \times CF \tag{13}$$

$$SA = 239 \times H^{0.417} \times BW^{0.517} \tag{14}$$

$$HQ_{dermal} = \frac{Intake_{dermal}}{RfD_{dermal}} \tag{15}$$

$$RfD_{dermal} = RfD_{oral} \times ABS_{gi} \tag{16}$$

where $Intake_{dermal}$ and $EV$ are the average daily exposure dosage(mg/(kg·d)) and frequency (1/d) by dermal intake, respectively. In this study, the value of $EV$ is 1, which assumes that all people in Xinzhou Basin are exposed to polluted water in various ways every day. $DA$ and $SA$ represent the exposure dosage (mg/cm$^2$) and exposed skin surface area (cm$^2$) of each individual event, respectively. $K$ and $t$ are the skin permeability coefficient (cm/h) and the contact duration (h/d), which are assigned with 0.001 and 0.4, respectively. $CF$ is a conversion factor and equals 0.001. $H$ represents the average height, which is 165.3 cm for men, 153.4 cm for women and 99.4 cm for children. $HQ_{dermal}$ and $RfD_{dermal}$ are the hazard quotient and reference dosage (mg/(kg·d)) of noncarcinogenic contaminants by dermal contact, respectively. $ABS_{gi}$ is the gastrointestinal absorption factor with a value of 1 [53].

The calculation parameters of different exposure pathways in the model are listed in Tables 2 and 3. The total noncarcinogenic risk is expressed as the hazard index (*HI*). A

value of *HI* < 1 means an acceptable noncarcinogenic risk, while *HQ* > 1 represents a high potential health risk, which is unacceptable for residents [53].

$$HI_i = HQ_{\text{oral}} + HQ_{\text{dermal}} \tag{17}$$

$$HI_{total} = \sum_{i=1}^{n} HI_i \tag{18}$$

where $HI_i$ is the hazard index of noncarcinogenic pollutant *i*, and $HI_{total}$ is the total hazard index of all noncarcinogenic pollutants concerned.

**Table 2.** Description and value of calculated parameters for noncarcinogenic risk by oral and dermal exposure.

| Parameter | Mean | Unit | Men | Women | Children |
|---|---|---|---|---|---|
| C | The contaminant concentration | mg/L | - | - | - |
| IR | Intake rate | L/day | 1.5 | 1.5 | 0.7 |
| EF | Exposure frequency | Days/year | 365 | 365 | 365 |
| ED | Exposure duration | Years | 30 | 30 | 12 |
| BW | Body weight | kg | 70 | 55 | 15 |
| AT | Average time | Days | $30 \times 365$ | $30 \times 365$ | $12 \times 365$ |
| EV | Daily exposure frequency | - | 1 | 1 | 1 |
| K | Permeability coefficient | cm/h | 0.001 | 0.001 | 0.001 |
| t | Exposure time | h/day | 0.4 | 0.4 | 0.4 |
| CF | Conversion factor | - | 0.001 | 0.001 | 0.001 |
| H | Average resident height | cm | 165.3 | 153.4 | 99.4 |

**Table 3.** Reference dose of noncarcinogens $RfD_i$ ($mg/kg \cdot d^{-1}$) and carcinogenic intensity coefficients of chemical carcinogens $q_i$ ($mg/kg \cdot d^{-1}$).

| Exposure Pathway | Noncarcinogens | $NO_3^-$ | $NO_2^-$ | $F^-$ |
|---|---|---|---|---|
| Direct ingestion | $RfD_i$ | 1.6 | 0.1 | 0.04 |
| Dermal absorption | $RfD_i$ | $1 \times 10^{-3}$ | $1 \times 10^{-3}$ | $1 \times 10^{-3}$ |

## 3. Results and Analysis

### 3.1. Groundwater Chemistry

#### 3.1.1. Hydrochemical Parameters

The statistical data of hydrochemical parameters of 31 groundwater samples and standard permissible limits for drink water [37] were listed in Table 4. The pH values varied from 7.60 to 8.38, which indicated that the groundwater was slightly alkaline in the study region. The values of TDS ranged between 211.45 and 904.07. TH was varied from 107.59 to 470.38 mg/L. According to the Chinese standard for drinking water quality, all the samples were within the allowable limit of TDS (1000 mg/L), while only one sample was more than the permissible limit of 450 mg/L for TH [37].

The hydrochemistry of groundwater depends on the content of major ions [55]. The average mass concentration of cations in descending order is $Ca^{2+} > Na^+ > Mg^{2+} > K^+$ and the anion is $HCO_3^- > SO_4^{2-} > Cl^- > NO_3^- > F^- > NO_2^-$. The type of groundwater was mainly Ca-HCO_3. The $Na^+$ in the groundwater was varied from 8.7 to 204.3 mg/L, with an average of 47.9 mg/L, and only one sample exceeded the permissible limit for 200 mg/L. The $K^+$ content is usually low in groundwater and mainly comes from feldspar or fertilizer. $K^+$ ranged between 0.5 and 5.7 mg/L, with a mean value of 2.0 mg/L. $Mg^{2+}$ and $Ca^{2+}$ are mainly from the dissolution of carbonate [56]. They are vital to human health, but they may have harmful effects on the human body at high concentrations. $Mg^{2+}$ ranged from 7.30 to 54.72 mg/L, with an average of 19.18 mg/L. $Ca^{2+}$ ranged between 31.06 and 98.20 mg/L, and its average was 60.02 mg/L. According to the mean value, $Ca^{2+}$ is significantly higher than $Na^+$, $Mg^{2+}$, and $K^+$. For the main anions, $SO_4^{2-}$ ranged from 2.40 to 355.42 mg/L; $Cl^-$ was ranged between 8.91 and 121.13 mg/L; $HCO_3^-$ ranged between

160.06 and 321.96 mg/L. According to China's drinking water standards, three samples exceeded the acceptable limit of $SO_4^{2-}$ (250 mg/L), while all the samples were within the limit of $Cl^-$ (250 mg/L) [37]. There were more $HCO_3^-$ anions than $SO_4^{2-}$ and $Cl^-$ anions in terms of their average content.

**Table 4.** The statistical data of hydrochemical parameters along with standard permissible limits for drink water in the study area.

| Sample | Unit | Number | Permissible Limit | Max | Min | Mean | SD | CV | National Standard [37] | Exceeding Standard |
|---|---|---|---|---|---|---|---|---|---|---|
| $Ca^{2+}$ | mg/L | 31 | 200 | 98.20 | 31.06 | 60.02 | 16.33 | 0.27 | - | - |
| $Mg^{2+}$ | mg/L | 31 | 50 | 54.72 | 7.30 | 19.18 | 9.57 | 0.50 | - | - |
| $K^+$ | mg/L | 31 | - | 5.70 | 0.50 | 2.05 | 1.22 | 0.60 | - | - |
| $Na^+$ | mg/L | 31 | 200 | 204.30 | 8.70 | 47.94 | 47.58 | 0.99 | 200 | 1 |
| $SO_4^{2-}$ | mg/L | 31 | 250 | 355.42 | 2.40 | 75.38 | 85.06 | 1.13 | 250 | 3 |
| $Cl^-$ | mg/L | 31 | 250 | 121.13 | 8.91 | 28.92 | 27.83 | 0.96 | 250 | 0 |
| $HCO_3^-$ | mg/L | 31 | 600 | 321.96 | 160.06 | 241.94 | 36.16 | 0.15 | - | - |
| $NO_3^-$ | mg/L | 31 | 20 | 43.60 | 1.00 | 12.96 | 9.37 | 0.72 | 20 | 5 |
| $NO_2^-$ | mg/L | 31 | 0.02 | 1.22 | 0.00 | 0.06 | 0.21 | 3.69 | 0.02 | 7 |
| PH | - | 31 | 6.5~8.5 | 8.38 | 7.60 | - | - | - | 6.5~8.5 | 0 |
| $F^-$ | mg/L | 31 | 1 | 3.50 | 0.00 | 0.60 | 0.74 | 1.24 | 1 | 3 |
| TH | mg/L | 31 | 450 | 470.38 | 107.59 | 228.81 | 66.49 | 0.29 | 450 | 1 |
| TDS | mg/L | 31 | 1000 | 904.07 | 211.45 | 374.10 | 163.47 | 0.44 | 1000 | 0 |
| COD | mg/L | 31 | 3 | 1.35 | 4.26 | 1.90 | 0.52 | 0.27 | 3 | 2 |
| SAR | - | 31 | - | 5.27 | 0.27 | 1.42 | 1.40 | 0.99 | - | - |
| RSC | - | 31 | - | 1.91 | −5.05 | −0.44 | 1.25 | −2.88 | - | - |

SD standards deviation, CV means Coefficient of Variation, "-" indicates no value.

Xinzhou Basin is an agricultural area where fertilizers and pesticides are widely used, so it is necessary to describe the content of nitrate and nitrite. As shown in Table 4, the $NO_3^-$ content ranged from 1.00 mg/L to 43.60 mg/L, with an average value of 12.96 mg/L. $NO_2^-$ was varied from 0 to 1.22 mg/L, with an average of 0.06 mg/L. In terms of the Chinese drinking water standard, 16.13% of the samples exceeded the prescribed limit of $NO_3^-$ (20 mg/L), while 22.58% of the samples exceeded the prescribed limit of $NO_2^-$ (0.02 mg/L).

It is also essential to pay attention to the fluorine content in groundwater since the residents in the study area have been suffering from fluorosis. $F^-$ is beneficial at low concentrations but toxic at high concentrations in drinking water [31,43]. In the study area, $F^-$ was varied from 0 to 3.5 mg/L with an average value of 0.6 mg/L, and the content in three samples surpassed the Chinese standard allowable limit of 1.0 mg/L for drinking water. Generally, the dissolution of fluorine-containing minerals such as amphibole and biotite is the primary source of $F^-$ in groundwater [50].

3.1.2. Types of Groundwater Based on Hydrochemical Characteristics

Piper diagram [57] is a simple and effective method to characterize and classify hydrochemistry according to the content of primary ions in groundwater samples [9,58]. As indicated in Figure 2, most of the samples fall in zone A, B or D of the triangle on the left, meaning that the cations in groundwater are mainly non-dominant, calcium and/or sodium type. The anion triangle on the right shows that 83.9% of the samples fall in zone E. The rest fall in zone B or zone C, indicating that the groundwater anion type is mainly bicarbonate type, followed by non-dominant type or chloride type.

However, almost all samples fall in zones II and IV, demonstrating that shallow groundwater in Xinzhou Basin mainly belongs to the Ca·Mg-HCO$_3$ type, Na-SO$_4$·Cl type and/or Ca·Mg-SO$_4$·Cl type. The samples were primarily distributed in zones 1, 3 and 5, indicating that almost all groundwater samples were Ca-HCO$_3$ type, alkaline earth exceeded alkalis and weak acids exceeded strong acids.

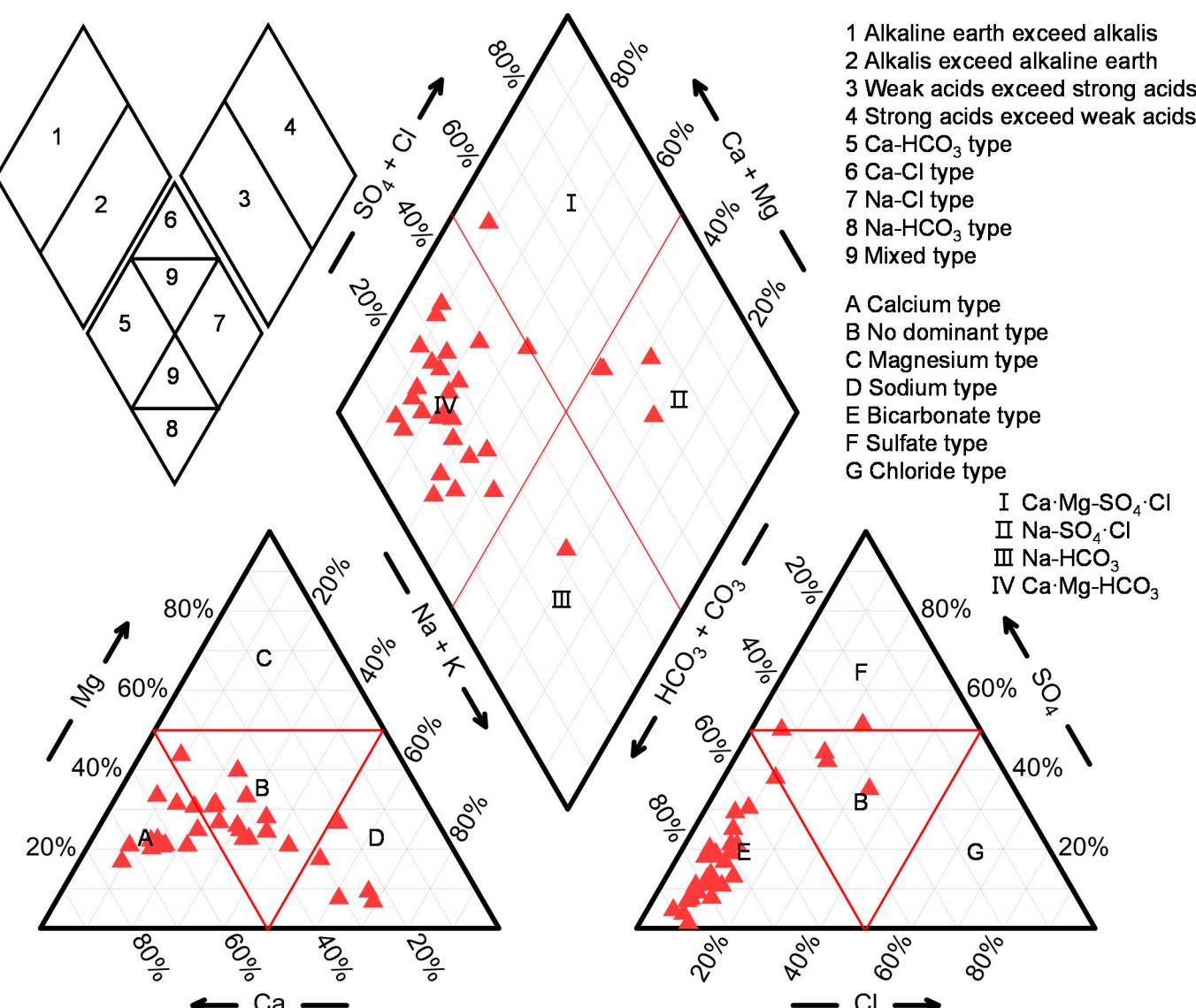

**Figure 2.** Piper diagram of the groundwater in the study area.

### 3.1.3. Groundwater Evolution Mechanisms

Gibbs diagrams are helpful for the rapid identification of the evolution mechanism of surface water [59], but now, they are also widely applied in groundwater studies [9,13]. Generally, the natural factors that control the chemical characteristics of groundwater include rainfall, evaporation and water–rock interaction.

Figure 3 shows that the $Na^+/(Na^++Ca^{2+})$ of most samples was less than or close to 0.5, the $Cl^-/(Cl^-+HCO_3^-)$ of all samples was less than 0.5, and the TDS values of all samples varied between 100 and 1000 mg/L. All samples were plotted in the rock dominated area, indicating that water–rock interactions and rock weathering were the main factors controlling the chemical characteristics of groundwater in Xinzhou Basin. Han et al. [35] reached a similar conclusion through isotope analysis in the study area. Groundwater in Xinzhou Basin flows in porous medium and its interaction with the aquifer medium controlled the chemical characteristics of groundwater. Intense rock weathering and water–rock interaction increase the dissolution of fluorine-bearing minerals, which may lead to higher concentrations of fluoride in groundwater in some areas of Xinzhou Basin.

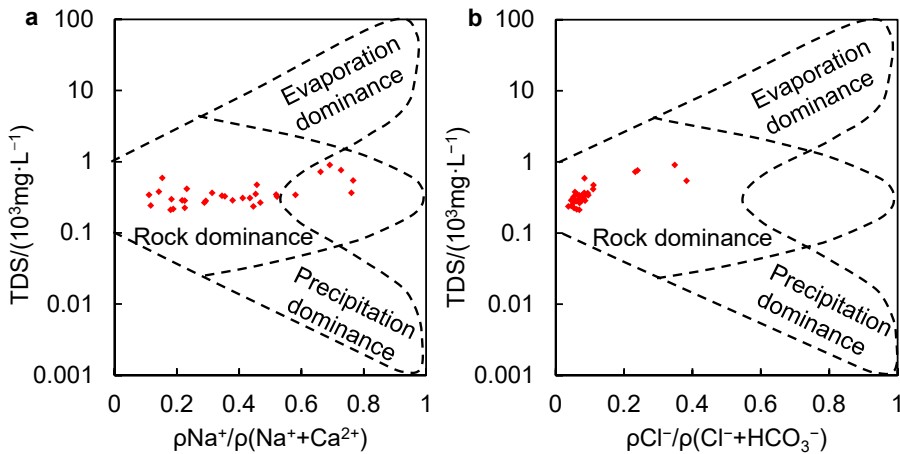

**Figure 3.** Gibbs diagrams of groundwater samples. (**a**) $\rho Na^+ / \rho(Na^+ + Ca^+)$; (**b**) $\rho Cl^- / \rho(Cl^- + HCO_3^-)$.

*3.2. Groundwater Quality Assessment*

3.2.1. Groundwater Quality Assessment for Drinking

The EWQI method was used to assess the groundwater quality of Xinzhou Basin for drinking, and $Cl^-$, $SO_4^{2-}$, $NO_3^-$, $NO_2^-$, $F^-$, COD, TH and TDS were considered. The Chinese drinking water quality standards were used as the standard limit for groundwater quality evaluation in the Xinzhou Basin. The groundwater quality level can be determined according to the distribution range of the EWQI value. If the EWQI is less than 25, it is defined as excellent quality; if it varies from 25 to 50, it is good quality; if it is between 50 and 100, the quality is medium; if it ranges between 100 and 150, it means poor quality; if it was larger than 150, the quality is extremely poor. When the quality is poor or extremely poor, the groundwater is not suitable for drinking, as in Table 1.

The results showed that the EWQI values varied between 58.37 and 246.23, with an average value of 103.07. Hence there was no excellent or good quality groundwater (grades 1 and 2) in Xinzhou Basin. Among 31 samples, 67.74% of groundwater samples were of medium quality and suitable for drinking purposes according to the classifications of EWQI. The quality of six (19.35%) and four (12.9%) samples were poor and extremely poor, respectively, which were considered unfit for drinking (grades 4 and 5). In Xinzhou Basin, the shallow groundwater in most regions was of medium quality and can be drinkable after simple treatment. The samples with high EWQI value were mainly distributed in Dingxiang County and Xinfu District in the south of Xinzhou Basin and Fanshi County in the northeast of Xinzhou Basin. Dingxiang County and Xinfu District were industrial and densely populated areas, and the groundwater in these areas may be mainly polluted by industrial wastewater. Fanshi County is known for planting rice, wheat and vegetables, and agricultural activities may be the main cause of groundwater pollution. The shallow groundwater body in the Xinzhou Basin can directly receive multiple replenishments. The surface water and groundwater in the basin have a relatively close hydraulic connection, easily polluted by agricultural, industrial or domestic sewage sources. The Hutuo River, which is the largest river in the region, accumulates pollutants due to seasonal flow changes and domestic sewage discharge, which affects the pollutant content of groundwater.

Nitrogen ($NO_3^-$ and/or $NO_2^-$) contaminated samples accounted for 32.26%, which were mainly distributed in the northeastern part of Xinzhou Basin, indicating that the study area was seriously polluted by nitrogen. Nitrogen is a useful indicator of agricultural pollution, mainly affected by agricultural production activities [56]. Large quantities of nitrogen-containing fertilizers were used to increase crop yields. A large amount of animal dung and sewage discharge were discharged without treatment. Nitrates and nitrites present in them increased the nitrogen content of groundwater through irrigation and forest filtration by rainfall. Residents of Fanshi County mainly rely on groundwater for drinking. If residents drink groundwater polluted by nitrate and nitrite for a long time,

it will have an adverse effect on their health, such as birth defects, methemoglobinemia, gastric cancer and other diseases [60,61].

Fluorine pollution is another major problem due to fluorine poisoning among residents in recent decades. Fluorine widely distributed in many fluoride–containing minerals such as amphibole, biotite and fluorite [6,31]. Fluorinated minerals are easily dissolved in an alkaline environment [62]. The concentration of fluoride in groundwater is not only affected by natural factors but also by human activities, such as coal combustion, brick kiln processing and aluminum smelting, and can also release fluoride into the environment, where it is deposited and enters groundwater [27,63]. The water sample with the highest fluoride ion content was situated in the central part of the Xinzhou Basin. The high fluoride concentration in this area may be due to the insufficient intensity of water circulation and easier evaporation and concentration of groundwater, resulting in fluoride enrichment. Although fluoride is beneficial at low levels, high fluoride concentrations pose a non-carcinogenic risk to humans, leading to health problems such as bone deformities and dental fluorosis [64]. The higher the fluoride concentration in drinking water, the greater the impact on human health.

### 3.2.2. Groundwater Quality Assessment for Irrigation

Besides, groundwater is also the primary source of agricultural irrigation water in Xinzhou Basin. When groundwater is used for irrigation, salinity hazards and alkali hazards must be considered [65,66]. Hence, the evaluation of groundwater quality for irrigation is of great significance to ensure the health of soil and plants.

The sodium adsorption ratio (SAR) and the soluble sodium percentage (%Na) are commonly used indicators to quantify potential sodium hazards in agricultural soil due to irrigation [25]. The high concentration of $Na^+$ in irrigation water may increase the osmotic pressure of the soil, reduce the permeability of the soil, limit the circulation of water and air in the roots of plants and affect the growth of plants. In addition, when a large amount of $Na^+$ is adsorbed on the soil particles, it causes the soil particles to disperse, resulting in decreased production due to difficulty in cultivation [66]. The residual sodium carbonate (RSC) can be used as an indicator of alkalinity damage in the soil to characterize the potential of water to remove $Ca^{2+}$ and $Mg^{2+}$ from soil solutions. If the RSC value of irrigation water is too high, it will limit the air and water flow through soil pores, leading to soil salinization and consolidation [65]. RSC indicates the potential of water to remove $Ca^{2+}$ and $Mg^{2+}$ from the soil solution. High RSC values in irrigation water may lead to salinization and solidification of agricultural soils [65]. In this study, SAR, %Na and RSC were used to evaluate groundwater quality for irrigation, and the formulas are listed below.

$$SAR = \frac{Na^+}{\sqrt{\frac{Ca^{2+}+Mg^{2+}}{2}}} \tag{19}$$

$$\%Na = \frac{Na^+}{Ca^{2+} + Mg^{2+} + Na^+ + K^+} \times 100 \tag{20}$$

$$RSC = \left(CO_3^{2-} + HCO_3^-\right) - \left(Ca^{2+} + Mg^{2+}\right) \tag{21}$$

If the SAR value of irrigation water is less than 10, the quality of irrigation water is excellent; the quality is considered good when the SAR is between 10 and 18. Water is deemed acceptable for irrigation when SAR is between 18 and 26; however, if the SAR value exceeds 26, the groundwater is not suitable for irrigation [67]. As listed in Table 2, the SAR value was between 0.27 and 5.27, with an average value of 1.42, indicating that all groundwater in Xinzhou Basin was excellent for irrigation.

In terms of %Na, if %Na is less than 30%, it means that the groundwater is suitable for irrigation; if %Na is in the range of 30~60%, it is acceptable; and if the value of %Na exceeds 60%, the groundwater is not suitable for irrigation [56]. As shown in Table 2, 28 (90.32%) samples were suitable for irrigation in terms of the value of %Na, while the remaining

samples were slightly above the acceptable limit (60%) for irrigation water. The average value of %Na is 27.26, which is less than 60%. This means that irrigation water quality can be improved in some areas by mixing groundwater from several sites, thus increasing the utilization of groundwater. Similarly, if the RSC value is less than 1.25 meq/L, the groundwater is suitable for irrigation; if the RSC value is greater than 2.5 meq/L, the water is not suitable for irrigation; if the RSC value of irrigation water is in the range of 1.25~2.5 meq/L, it is barely acceptable [56]. In this study, the RSC range was −5.05 to 1.91, indicating that all the groundwater in Xinzhou Basin was suitable for irrigation.

According to SAR, RSC and %Na, 90.32% of groundwater samples were suitable for irrigation, while %Na of the remaining samples exceeded the limit. However, the high salinity of groundwater in some areas of Xinzhou Basin was not conducive to the growth of crops. Some appropriate measures need to be taken, such as mixing various water sources, to eliminate the salt damage caused by irrigation water to the land and plants.

### 3.3. Human Health Risk Assessment

The above results showed that the groundwater in Xinzhou Basin was mainly contaminated by fluoride, nitrate and nitrite. In addition, 32.26% of the samples belonged to groundwater types classified as having a poor drinking quality or extremely poor quality. Therefore, it was essential to evaluate the potential health risks of fluoride, nitrate and nitrite in groundwater to residents of Xinzhou. The HHRA model is an effective method to assess the noncarcinogenic health risks of different groups (such as men, women and children) in Xinzhou Basin [5,68]. The calculation results of noncarcinogenic health risks for men, women and children in the study area through the oral intake and dermal contact are presented in Table 5.

For men, the $HQ_{oral}$ value ranged from 0.02 to 2.14, with an average value of 0.51; the $HQ_{dermal}$ value was less than $HQ_{oral}$ and ranged from $8.84 \times 10^{-5}$ to $1.03 \times 10^{-2}$, with a mean value of $2.45 \times 10^{-2}$. This indicated that the oral intake of polluted water was the primary exposure pathway to noncarcinogenic risk. Moreover, the value of $HI_{total}$ for men ranged from 0.02 to 2.15. The $HQ_{oral}$ and $HI_{total}$ values of 3 (9.68%) samples were greater than 1, indicating that these samples may pose noncarcinogenic risks to men if they drink contaminated groundwater in Xinzhou Basin. Similarly, the $HQ_{oral}$ were 0.02 to 2.72 for women and 0.04 to 4.66 for children; the $HQ_{dermal}$ values were $9.62 \times 10^{-5}$~$1.12 \times 10^{-2}$ and $1.5 \times 10^{-5}$~$1.76 \times 10^{-3}$, respectively. The results showed that the $HI_{total}$ values for women and children were 0.02~2.74 and 0.04~4.66, respectively. The $HQ_{oral}$ values for women and children exceeded the standard in 4 (12.9%) and 12 (38.71%) samples, respectively.

The health risk for children was approximately 2.18 times that for men and 1.71 times that for women, indicating that children were more sensitive to groundwater pollution. The $HQ_{oral}$, $HQ_{dermal}$ and $HI_{total}$ values of women and children were higher than those of men, which may be related to the lower body weight of women and children than men.

The contribution of pollutants to health risks varies. $F^-$ has the largest contribution to noncarcinogenic risks (63.4%), followed by $NO_3^-$ (34.15%). The contribution of other pollutants to noncarcinogenic risks was less than 2.45%, indicating that $F^-$ and $NO_3^-$ may be the main factors threatening the health of Xinzhou residents. The impact of various pollutants on the potential noncarcinogenic health risks of residents increased in the order of $NO_2^- < NO_3^- < F^-$.

**Table 5.** Assessment results of health risks through drinking water intake and dermal contact.

| Sample | HQ$_{oral}$ | | | HQ$_{dermal}$ | | | HI$_{total}$ | | |
|---|---|---|---|---|---|---|---|---|---|
| | Men | Women | Children | Men | Women | Children | Men | Women | Children |
| S1 | 0.29 | 0.37 | 0.63 | $1.39 \times 10^{-3}$ | $1.51 \times 10^{-3}$ | $2.36 \times 10^{-4}$ | 0.29 | 0.37 | 0.63 |
| S2 | 0.44 | 0.56 | 0.96 | $2.13 \times 10^{-3}$ | $2.32 \times 10^{-3}$ | $3.63 \times 10^{-4}$ | 0.44 | 0.57 | 0.96 |
| S3 | 0.53 | 0.68 | 1.16 | $2.57 \times 10^{-3}$ | $2.80 \times 10^{-3}$ | $4.37 \times 10^{-4}$ | 0.53 | 0.68 | 1.16 |
| S4 | 2.14 | 2.72 | 4.66 | $1.03 \times 10^{-2}$ | $1.12 \times 10^{-2}$ | $1.76 \times 10^{-3}$ | 2.15 | 2.74 | 4.66 |
| S5 | 0.28 | 0.36 | 0.62 | $1.37 \times 10^{-3}$ | $1.50 \times 10^{-3}$ | $2.34 \times 10^{-4}$ | 0.29 | 0.36 | 0.62 |
| S6 | 0.52 | 0.66 | 1.13 | $2.51 \times 10^{-3}$ | $2.73 \times 10^{-3}$ | $4.27 \times 10^{-4}$ | 0.52 | 0.66 | 1.13 |
| S7 | 0.55 | 0.71 | 1.21 | $2.68 \times 10^{-3}$ | $2.91 \times 10^{-3}$ | $4.55 \times 10^{-4}$ | 0.56 | 0.71 | 1.21 |
| S8 | 0.44 | 0.57 | 0.97 | $2.14 \times 10^{-3}$ | $2.34 \times 10^{-3}$ | $3.65 \times 10^{-4}$ | 0.45 | 0.57 | 0.97 |
| S9 | 1.24 | 1.57 | 2.69 | $5.96 \times 10^{-3}$ | $6.49 \times 10^{-3}$ | $1.01 \times 10^{-3}$ | 1.24 | 1.58 | 2.69 |
| S10 | 1.53 | 1.94 | 3.32 | $7.36 \times 10^{-3}$ | $8.02 \times 10^{-3}$ | $1.25 \times 10^{-3}$ | 1.53 | 1.95 | 3.33 |
| S11 | 0.14 | 0.18 | 0.31 | $6.84 \times 10^{-4}$ | $7.45 \times 10^{-4}$ | $1.16 \times 10^{-4}$ | 0.14 | 0.18 | 0.31 |
| S12 | 0.47 | 0.59 | 1.01 | $2.24 \times 10^{-3}$ | $2.44 \times 10^{-3}$ | $3.82 \times 10^{-4}$ | 0.47 | 0.59 | 1.01 |
| S13 | 0.40 | 0.51 | 0.87 | $1.93 \times 10^{-3}$ | $2.10 \times 10^{-3}$ | $3.28 \times 10^{-4}$ | 0.40 | 0.51 | 0.87 |
| S14 | 0.27 | 0.35 | 0.60 | $1.32 \times 10^{-3}$ | $1.44 \times 10^{-3}$ | $2.24 \times 10^{-4}$ | 0.27 | 0.35 | 0.60 |
| S15 | 0.75 | 0.95 | 1.63 | $3.61 \times 10^{-3}$ | $3.93 \times 10^{-3}$ | $6.14 \times 10^{-4}$ | 0.75 | 0.96 | 1.63 |
| S16 | 0.79 | 1.01 | 1.73 | $3.82 \times 10^{-3}$ | $4.16 \times 10^{-3}$ | $6.50 \times 10^{-4}$ | 0.80 | 1.01 | 1.73 |
| S17 | 0.29 | 0.37 | 0.63 | $1.39 \times 10^{-3}$ | $1.52 \times 10^{-3}$ | $2.37 \times 10^{-4}$ | 0.29 | 0.37 | 0.63 |
| S18 | 0.30 | 0.39 | 0.66 | $1.47 \times 10^{-3}$ | $1.60 \times 10^{-3}$ | $2.50 \times 10^{-4}$ | 0.31 | 0.39 | 0.66 |
| S19 | 0.18 | 0.23 | 0.40 | $8.83 \times 10^{-4}$ | $9.62 \times 10^{-4}$ | $1.50 \times 10^{-4}$ | 0.18 | 0.23 | 0.40 |
| S20 | 0.27 | 0.34 | 0.59 | $1.30 \times 10^{-3}$ | $1.41 \times 10^{-3}$ | $2.21 \times 10^{-4}$ | 0.27 | 0.34 | 0.59 |
| S21 | 0.44 | 0.57 | 0.97 | $2.14 \times 10^{-3}$ | $2.33 \times 10^{-3}$ | $3.65 \times 10^{-4}$ | 0.45 | 0.57 | 0.97 |
| S22 | 0.36 | 0.46 | 0.78 | $1.74 \times 10^{-3}$ | $1.89 \times 10^{-3}$ | $2.96 \times 10^{-4}$ | 0.36 | 0.46 | 0.78 |
| S23 | 0.63 | 0.81 | 1.38 | $3.05 \times 10^{-3}$ | $3.33 \times 10^{-3}$ | $5.20 \times 10^{-4}$ | 0.64 | 0.81 | 1.38 |
| S24 | 0.35 | 0.45 | 0.76 | $1.69 \times 10^{-3}$ | $1.84 \times 10^{-3}$ | $2.88 \times 10^{-4}$ | 0.35 | 0.45 | 0.76 |
| S25 | 0.02 | 0.02 | 0.04 | $8.84 \times 10^{-5}$ | $9.62 \times 10^{-5}$ | $1.50 \times 10^{-5}$ | 0.02 | 0.02 | 0.04 |
| S26 | 0.13 | 0.17 | 0.29 | $6.41 \times 10^{-4}$ | $6.98 \times 10^{-4}$ | $1.09 \times 10^{-4}$ | 0.13 | 0.17 | 0.29 |
| S27 | 0.65 | 0.83 | 1.43 | $3.16 \times 10^{-3}$ | $3.44 \times 10^{-3}$ | $5.37 \times 10^{-4}$ | 0.66 | 0.84 | 1.43 |
| S28 | 0.16 | 0.20 | 0.34 | $7.59 \times 10^{-4}$ | $8.27 \times 10^{-4}$ | $1.29 \times 10^{-4}$ | 0.16 | 0.20 | 0.34 |
| S29 | 0.49 | 0.62 | 1.06 | $2.34 \times 10^{-3}$ | $2.55 \times 10^{-3}$ | $3.99 \times 10^{-4}$ | 0.49 | 0.62 | 1.06 |
| S30 | 0.34 | 0.43 | 0.74 | $1.64 \times 10^{-3}$ | $1.78 \times 10^{-3}$ | $2.79 \times 10^{-4}$ | 0.34 | 0.43 | 0.74 |
| S31 | 0.34 | 0.43 | 0.74 | $1.64 \times 10^{-3}$ | $1.79 \times 10^{-3}$ | $2.79 \times 10^{-4}$ | 0.34 | 0.43 | 0.74 |
| min | 0.02 | 0.02 | 0.04 | $8.84 \times 10^{-5}$ | $9.62 \times 10^{-5}$ | $1.50 \times 10^{-5}$ | 0.02 | 0.02 | 0.04 |
| max | 2.14 | 2.72 | 4.66 | $1.03 \times 10^{-2}$ | $1.12 \times 10^{-2}$ | $1.76 \times 10^{-3}$ | 2.15 | 2.74 | 4.66 |
| Exceed number | 3 | 4 | 12 | 0 | 0 | 0 | 3 | 4 | 12 |

## 4. Discussion

Groundwater was essential to ensure the health of humans and crops in Xinzhou Basin. However, this study showed that the groundwater used for drinking and irrigation in parts of the study area posed a potential noncarcinogenic risk to human health. Groundwater, especially in agricultural and industrial areas, was deteriorating because of a lack of sewage treatment procedures [62]. Therefore, it was necessary to strengthen the quality management of groundwater in Xinzhou Basin, take some measures to improve groundwater quality, reduce the potential health risks of groundwater to human beings and realize the sustainable utilization and management of groundwater.

Moreover, nitrite, nitrate and fluoride contamination and health risk assessments have been studied by many researchers worldwide [8,9,69]. Similar conclusions have been found in different regions of China such as Guanzhong Basin [60], Ordos Basin [70] and Weining Plain [50], and Western Gujarat, India [71], and Sanandaj city, Iran [72]. They found that children are more likely to be exposed to potential health risks because they weigh less than adults. Drinking groundwater with high concentrations of nitrogen and fluoride can lead to blue baby syndrome/methemoglobinemia [73], stomach cancer [60] and dental and skeletal fluorosis [69]. In this study, nitrate and nitrite are the primary pollutants in the

groundwater of Xinzhou Basin. Nitrates from overuse of nitrogen-containing fertilizers, domestic sewage discharge, landfill leachate and fecal leakage seep into groundwater through irrigation and rainfall [74]. In addition, fluoride-bearing minerals such as mica and hornblende are widely distributed in the mountain strata around the basin. Under the influence of rock weathering and water-rock interaction, the fluorine in the fluoride-bearing minerals is dissolved in the groundwater and enriched in the basin, which increases the concentration of fluoride in the groundwater [31]. This may also be an important cause of fluorosis among Xinzhou basin residents.

Besides, there might be uncertainties in the health risk assessment used in this study. The selection of models and parameters might introduce some uncertainties into the health risk assessment. For example, the parameters of BW, AT and EV in the model were based on statistical averages [50]. However, different individuals may possess different parameter values, and different intake rates lead to variable risks. Furthermore, other toxic contaminations may pose a threat to human health, such as pesticides and heavy metals, which were not tested in groundwater samples and were not considered in the assessment of potential health risks to Xinzhou residents [11], will also cause deviations in the result. Despite these uncertain factors, the research results are still valid and meaningful and can provide information for management to improve groundwater conditions.

## 5. Conclusions

A case study investigating groundwater pollution and potential risks to human health is presented in this paper. Sampling and collecting of groundwater in Xinzhou Basin were first carried out. The hydrochemical characteristics and distributions of the groundwater samples were carefully analyzed. The hydrochemistry types and evolutionary mechanisms of groundwater were analyzed, and the groundwater quality and noncarcinogenic human health risks were evaluated. The investigation results indicate the following:

(1)  $Ca$-$HCO_3$ and $Ca \cdot Mg$-$HCO_3$ were the dominant water types. The hydrochemical characteristics of groundwater were mainly governed by rock weathering and water–rock interactions.

(2)  Based on the EWQI classifications, 67.74% of the groundwater samples were classified as medium quality and drinkable. According to the values of SAR, RSC and %Na, 90.32% of the samples were suitable for irrigation, while the remaining samples were unfit for irrigation due to the high salinity of groundwater.

(3)  $NO_3^-$, $NO_2^-$ and $F^-$ were the main contaminants of groundwater in the study area. The noncarcinogenic risks of some groundwater samples exceed acceptable levels. $F^-$ and $NO_3^-$ were the main contaminants contributing to the total noncarcinogenic risk. The order of contaminant contribution to noncarcinogenic risk was $F^- > NO_3^- > NO_2^-$.

**Author Contributions:** Conceptualization, F.C. and L.Y.; methodology, F.C. and G.M.; sampling, Y.S.; analysis, F.C., L.Y. and G.M.; investigation, F.C., L.Y. and Y.S.; writing—original draft preparation, F.C., F.X. and Z.D.; writing—review and editing, L.Y., G.M., F.X., Z.D. and Y.S.; visualization, F.C., F.X. and Z.D. All authors have read and agreed to the published version of the manuscript.

**Funding:** This research received no external funding.

**Institutional Review Board Statement:** Not applicable.

**Informed Consent Statement:** Not applicable.

**Data Availability Statement:** Not applicable.

**Acknowledgments:** We would like to thank the anonymous reviewers and the editor.

**Conflicts of Interest:** The authors declare no conflict of interest.

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
