# Peer review of "Groundwater Quality and Potential Human Health Risk Assessment for Drinking and Irrigation Purposes: A Case Study in the Semiarid Region of North China"

_water, doi:10.3390/w13060783_

Round 1

Reviewer 1 Report

The authors have evaluated the groundwater quality and the potential health risk from consuming groundwater in a part of North China. This is an important issue to be addressed. This manuscript presents an interesting study in this field and the findings might be useful to other studies aimed at similar topics. The methods are presented clearly enough to allow other researchers to repeat the study.  The results respond to the aim of the study. However, the manuscript reserves some scope for improvement. I would suggest minor revision must be carried out before the paper is accepted.

Specific comments:

-L13, groundwater pollution does not affect human health directly unless it is used without pre-treatment and for human use. This should be clarified.

- is there a specific reason for not using the conventional cation-anion water type and using the anion-cation style.

-L23, presence of certain parameters in groundwater does not imply that have potential health risk. This should be clearly explained. The presence of certain parameters above the permissible limits in groundwater which is used for drinking may cause potential risk to human health. There are several sentences in the manuscripts which have to rephrased for better understanding. Eg L36-37,

- Introduction is weak and does not bring out the novelty of the presented study and does not have any problem statements. What makes the study area special? What are the key features? Are there industrial zones? What are the specific contaminants in groundwater in the area? The authors study the groundwater quality with respect to drinking and irrigation use in the region. How much % of groundwater is used for water supply? How much percentage is surface water? What is the major surface water source? Are there no piped water distribution network by the government? If so, what is the source of this water? What is the demand? How much is provided by the government and how much is from private borewells? What % of water is used for settlements and what % for agriculture etc? There are several questions that a reader has while reading the introduction. I suggest that the authors rewrite the introduction considering all these questions and concentrating more on the water resources. Also include, what is the significance of this study in the study area? L159-167 should be explained in introduction.

- table 1, what is the criteria or reference for this EWQI classification?

- Did you filter (0.45 μm) the samples? This is important as you are interested in the dissolved parameters only. Did you acidify the samples (e.g. with HNO3) to stabilize cations in solution?

- how was the precision and accuracy of the methods ensured?

-L299, ‘standard limits’ please be specific, is it WHO standards or Chinese national standards?

-pH is a logarithmic parameter and it is not correct to calculate the arithmetic mean for pH. Please modify in table 4 and in text.

-table 4, include reference for the national standards

-section 3.2.1, there are too many paragraphs in this section with only 2-3 sentences. They can be combined.

-Authors have mentioned about health risks from consuming heavy metals dissolved in groundwater and I was expecting that some of these would also have been analysed in the study. It would be best to clarify these points in the methods section.

-L536-538, the major source for most ions in groundwater is mentioned as rock weathering and rock water interactions. There is no justifiable text for the source of nitrogen-species and F in groundwater. The contribution of agriculture or industries are not explained in the manuscript, yet is mentioned in the conclusion.

Author Response

Dear Reviewer,

We have made a point-by-point response to the comments and suggestions, including a detailed description of any requested or suggested revisions.

All the modifications and explanations in this revised version are listed in detail in the following “Response to the 1st Reviewer”.

We would deeply appreciate your consideration and helpful comments and suggestions.

Yours Sincerely,

Feifei Chen, Leihua Yao, Gang Mei, Yinsheng Shang, Fansheng Xiong, Zhenbin Ding

School of Engineering and Technology, China University of Geosciences (Beijing)

Email: [email protected] (Leihua Yao)

Reviewer 2 Report

Line 16: Specify the type of statistical methods used.

Lines 196 – 214 (steps 1 – step 6): These described the general steps for calculating the EWQI. However, it will be more appropriate to describe the methods used for this study instead.

Author Response

Dear Reviewer,

We have made a point-by-point response to the comments and suggestions, including a detailed description of any requested or suggested revisions.

All the modifications and explanations in this revised version are listed in detail in the following “Response to the 2nd Reviewer”.

We would deeply appreciate your consideration and helpful comments and suggestions.

Yours Sincerely,

Feifei Chen, Leihua Yao, Gang Mei, Yinsheng Shang, Fansheng Xiong, Zhenbin Ding

School of Engineering and Technology, China University of Geosciences (Beijing)

Email: [email protected] (Leihua Yao)

Reviewer 3 Report

The manuscript under the title "Groundwater Quality and Potential Human Health Risk Assessment for Drinking and Irrigation Purposes: A Case Study in the Semiarid Region of North China" is well written and interesting, however I have some comments to the Authors, see below:

Page 3 - I suggest moving Fig. 1 from the "Introduction" to  "Materials and Methods".

Page 9 - Please format Table 4 to fit within the margins of a standard A4 page. This can be achieved by taper individual columns. 

Page 10 - Please quote Figure 2 first and then plot it. Please change it. 

Page 8-15 - The "Results and discussion" lacks the results obtained by other researchers in the discussed topic. Please complete this. 

Author Response

Dear Reviewer,

We have made a point-by-point response to the comments and suggestions, including a detailed description of any requested or suggested revisions.

All the modifications and explanations in this revised version are listed in detail in the following “Response to the 3rd Reviewer”.

We would deeply appreciate your consideration and helpful comments and suggestions.

Yours Sincerely,

Feifei Chen, Leihua Yao, Gang Mei, Yinsheng Shang, Fansheng Xiong, Zhenbin Ding

School of Engineering and Technology, China University of Geosciences (Beijing)

Email: [email protected] (Leihua Yao)

Round 2

Reviewer 1 Report

Authors have significantly improved the manuscript based on the comments. 

Author Response

Dear Reviewer,

Thank you very much for your comments.

Yours Sincerely,

Feifei Chen, Leihua Yao, Gang Mei, Yinsheng Shang, Fansheng Xiong, Zhenbin Ding

School of Engineering and Technology, China University of Geosciences (Beijing)

Email: [email protected] (Leihua Yao)

Reviewer 3 Report

I have no other comments to the Authors.

Author Response

(The authors gave the same response as above.)
